# An Assessment of Regional Genetic Diversity of HIV-1

**DOI:** 10.3390/v17121568

**Published:** 2025-11-30

**Authors:** Anastasiia Antonova, Anna V. Kuznetsova, Anna I. Kuznetsova, Aleksei Mazus, Ekaterina Loifman, Liudmila Grigoreva, Denis Kleimenov, Evgeniia Bykonia, Dmitry Shcheblyakov, Irina Favorskaya, Andrei Pochtovyi, Elena Tsyganova, Inna Kulikova, Andrei Plutnitskii, Vladimir Gushchin, Aleksandr Gintsburg

**Affiliations:** 1Federal State Budget Institution “National Research Center for Epidemiology and Microbiology Named After the Honorary Academician N.F. Gamaleya”, The Ministry of Health of the Russian Federation, 123098 Moscow, Russia; 2Regional State Budgetary Healthcare Institution “Center for the Prevention and Control of AIDS and Infectious Diseases”, The Ministry of Health of the Khabarovsk Krai, 680031 Khabarovsk, Russia; 3Moscow City Center for AIDS Prevention and Control, 105275 Moscow, Russia; 4Department of Medical Genetics and Postgenomic Technologies, Sechenov First Moscow State Medical University, 119991 Moscow, Russia; 5Department of Emergency Medical Assistance Organisation and Health Risk Management, Ministry of Health of the Russian Federation, 127994 Moscow, Russia; 6Federal Medical Biophysical Center Named After A.I. Burnazyan, 123098 Moscow, Russia; 7Department of Virology, Lomonosov Moscow State University, 119234 Moscow, Russia; 8Department of Infectiology and Virology, Sechenov First Moscow State Medical University, 119991 Moscow, Russia

**Keywords:** HIV-1, genetic variants, genetic diversity, vaccine, cluster analysis, drug resistance mutations

## Abstract

This study aimed to assess the genetic diversity of HIV-1 in the Far Eastern Federal District (Russia) to implement effective anti-epidemic measures, including the development of an anti-HIV vaccine and the selection of optimal antigens. The first stage of the study included an analysis of HIV-1 nucleotide sequences obtained in Khabarovsk city from 2022 to 2024. The second stage of the study included an additional download of nucleotide sequences from the Los Alamos HIV Sequence Database for phylogenetic cluster analysis. Additionally, an analysis of drug resistance mutations was conducted. The results showed the following distribution of HIV-1 genetic variants: A6—72.15%, CRF63—10.13%, URFs—7.59%, C—5.06%, B—3.8%, and CRF157—1.27%. The phylogenetic cluster analysis revealed a statistically significant difference in the number of clusters depending on the genetic variant. Among drug resistance mutations (DRMs), those associated with nucleoside reverse transcriptase inhibitors (NRTIs) were the most frequently observed, accounting for 55.7% (95% CI: 44.75%—66.65%). The most commonly detected NRTI DRMs were A62V (43.04%) and M184V (13.92%). The results of this study highlight several important indicators for public health, particularly in the development of vaccines aimed at combating HIV infection.

## 1. Introduction

Currently, HIV infection remains a significant threat to public health worldwide, despite notable progress in the development and availability of antiretroviral therapy (ART). By the end of 2024, the global number of people living with HIV (PLWH) was 40.8 million, with 927689 individuals in the Russian Federation [1].

One of the key characteristics of HIV-1, which simultaneously poses a challenge to achieving success in combating the HIV pandemic globally, is the extremely high genetic and, consequently, antigenic variability [2]. The high frequency of mutations and the ability to recombine lead to the formation of many HIV-1 genovariants. To date, 10 “pure” HIV-1 subtypes and over 170 circulating recombinant forms (CRFs) have been identified [3]. Moreover, the differences between these HIV-1 genovariants can affect the effectiveness of HIV prevention and therapy, complicating the selection of optimal antigens and the development of a universal vaccine [2,4,5].

Currently, there is a steady increase in HIV-1 genetic diversity over time, both globally and regionally. Assessing this genetic diversity is critical to developing effective strategies to combat the pandemic [6]. This assessment helps identify predominant genetic variants in different geographic regions, enabling the selection of optimal antigens and evaluation of the effectiveness of existing treatment regimens within target populations [6,7].

The Russian Federation has several distinctive features that can influence the course of the epidemic and the effectiveness of anti-epidemic measures: Russia encompasses a vast territory where people of various age groups, nationalities, and cultural practices coexist, with differing lifestyles and mobility patterns. Additionally, the Russian Federation shares borders with 18 countries, contributing to high genetic diversity and the rapid spread of HIV-1 due to migration processes. The unique nature of the distribution of HIV-1 genetic variants in the territory of the Russian Federation is attributed to the “founder effect”—in the late 1990s, the virus of sub-subtype A6 entered the country through injecting drug users (IDUs) in the southern regions, and then quickly and widely spread throughout the country [8]. Presently, this variant accounts for 70–80% of the total diversity. Also, the following HIV-1 genovariants have been reported in Russia: A1, B, C, F1, G, CRF01, CRF02, CRF03, CRF11, CRF63, and unique forms. The distribution of HIV genetic variants is uneven across regions, and the Far Eastern Federal District (FEFD) exhibits high genetic diversity and a predominance of recombinant forms [9].

Thus, the Far Eastern Federal District is of particular interest for studying the HIV-1 genetic diversity due to several factors. Firstly, the FEFD borders areas that are epicentres of the spread of specific HIV-1 genetic variants, such as the CRF01 and CRF07 in the countries of Southeast Asia [10]. Migration and tourist activity of the population between the FEFD and neighbouring countries create conditions for the importation and spreading of new HIV-1 variants. Secondly, the FEDF has a high prevalence of drug addiction, which can also contribute to the active transmission of HIV-1 [11].

Also, prior research has indicated a significant and pronounced upward trend in HIV incidence in the FEFD, with an annual average growth rate (AAGR) of 8.14%. A similar upward trend was also observed in prevalence, escalating from 128.23 to 347.83 cases per 100,000 people during the 2011–2022 period. Despite a high ART coverage in the FEFD (92%), viral suppression rates among those infected were the lowest in Russia, reaching only 62% [12]. Consequently, such indicators may also influence the spread of drug-resistant variants of HIV-1.

The aim of this study is to assess the genetic diversity of HIV-1 in the Far Eastern Federal District using cluster analysis methods to identify predominant subtypes, recombinant forms, and potential transmission clusters. This is essential for the implementation of effective anti-epidemic measures, including the development of an anti-HIV vaccine and the selection of optimal antigens.

## 2. Materials and Methods

### 2.1. Study Design and Data Collection

In this study, we analysed the genetic diversity of HIV-1 in the Far Eastern Federal District of Russia, an area characterised by high genetic diversity, particularly due to its borders with countries that have different HIV-1 genetic compositions.

The first stage of the study included an analysis of HIV-1 nucleotide sequences obtained in the Federal District (Khabarovsk city) from 2022 to 2024 (n = 79).

The second stage of the study included an additional download of nucleotide sequences from the Los Alamos HIV Sequence Database for phylogenetic cluster analysis.

Additionally, an analysis of drug resistance mutations was conducted for the 79 patients mentioned earlier.

### 2.2. Sequence Analysis and Dataset

The material of the first stage of the study included sequences (n = 79), obtained from HIV-infected ART-treated patients at the “Centre for AIDS and Infectious Diseases Prevention and Control” of the Khabarovsk Region Ministry of Health (Russia) (hereinafter referred to as the AIDS Centre) in the period of 2022–2024. The total number of nucleotide sequences was 79. All sequences used in this study are available through GenBank via accession numbers: PP221634-PP221712.

Each nucleotide sequence was accompanied by patient demographic and epidemiological data collected through a standardised study according to national regulations, and all data were anonymised and coded to ensure confidentiality in accordance with the ethical standards of the Russian Federation.

Firstly, the HIV-1 *pol* gene region, encoding the PR-RT (2253–3369 bp according to the HXB2 strain, GenBank accession number: K03455), was sequenced from 79 patient samples using the commercial genotyping kits (the AmpliSens^®^ HIV-Resist-Seq (Central Research Institute of Epidemiology, Moscow, Russia)).

Then, multiple sequence alignments were generated using the ClustalW module within AliView v.1.27 [13]. Regions of ambiguity in the initial alignments were subjected to manual refinement. The length of the final alignment was 1101 bp, i.e., it covered the complete protease coding region and 268 amino acids of the reverse transcriptase (2253–3353 bp according to the HXB2 strain, GenBank accession number: K03455).

Determination of the HIV-1 genetic variant was carried out using specialised online programs: COMET HIV-1 [14], HIVdbProgram Sequence Analysis (version 9.8), presented on the website of Stanford University [15], and REGA HIV-1 Subtyping Tool (version 3.46) [16], according to the algorithm described in the previous study [9]. To investigate potential recombination events, sequences containing undetermined genetic variants were subjected to additional analysis using the jpHMM algorithm [17]. To confirm the genotyping results, phylogenetic analysis was performed. Well-defined, unique recombinant forms (based on jpHMM results) were preliminarily excluded from the phylogenetic analysis. HIV-1 subtype references were downloaded (https://www.hiv.lanl.gov/content/sequence/NEWALIGN/align.html, accessed on 4 August 2024) and further added to the analysis. Phylogenetic analyses were performed via the maximum likelihood method using IQ-TREE (version 2.0.3) with the following command-line arguments: iqtree -s [sequence alignment file] -m MFP -bb 1000. The best-fit nucleotide substitution model was determined automatically: GTR + F + R4 (chosen according to BIC). Phylogenetic tree visualisation and annotation were performed using the iTOL (Interactive Tree Of Life) v7 software [18].

For the second part of the study, nucleotide sequences of HIV-1 variants different from A6, which is the most widespread genetic variant in Russia, were selected. Among the sequences studied, these were the genetic variants: B, C, CRF63, and CRF157. Additional sequences were downloaded from the Los Alamos HIV Sequence Database (https://www.hiv.lanl.gov/content/index/, accessed on 4 August 2025) with following criteria: 1. nucleotide sequences were obtained in large cities (with a population of more than 100 thousand people, and the city of sample collection is exactly known) of the Far Eastern Federal District (included: Khabarovsk, Vladivostok, Blagoveshchensk, Yuzhno-Sakhalinsk, Nakhodka, Chita, Yakutsk, Ulan-Ude), 2. the *pol* gene coding region, 3. the genetic variant, which is exactly defined and different from A6.

To identify phylogenetically related sequences, a BLAST (version 2.2.30) search (https://www.hiv.lanl.gov/content/sequence/BASIC_BLAST/basic_blast.html, accessed on 4 August 2024) was performed on both the studied and downloaded sequences. Additionally, reference nucleotide sequences of different genetic variants and geographic clades were also added. After removing all duplicate sequences by “Patient Code”, “PAT id”, “Name”, and “Accession”, phylogenetic analysis was repeated as described previously. Additionally, background information for sequences was downloaded, including gender, age, route of infection, and country of infection. The total number of sequences downloaded from GenBank was 692 nucleotide sequences, 176 of which were from the Far East. Collection dates of sequences from the Far East ranged from 2012 to 2024.

### 2.3. Cluster Analysis

ClusterPicker v.1.2.3 software was used to identify potential transmission clusters between Russian non-A6 HIV-1 genetic variants and sequences from other regions and countries [19]. A total of 708 HIV-1 *pol* sequences were analysed. Phylogenetic clusters were defined as groups of sequences with an intra-cluster genetic distance threshold of ≤1.5% and 0.5% (0.015 and 0.005 nucleotide substitutions per site) and a bootstrap support value ≥ 90% [20].

### 2.4. Analysis of Drug Resistance Mutations

Analysis of drug resistance mutations (determination of DRMs to protease inhibitors (PIs), nucleoside reverse transcriptase inhibitors (NRTIs), and non-nucleoside reverse transcriptase inhibitors (NNRTIs)) was performed using the Sierra algorithm implemented in the HIVdb Program: Mutations Analysis Tool version 9.7 (Stanford University HIV Drug Resistance Database; https://hivdb.stanford.edu/hivdb/by-sequences/, accessed on 4 August 2024).

### 2.5. Statistical Analysis

Statistical analysis was performed to assess differences in demographic, epidemiological, and clinical characteristics of HIV-1 genetic variants (identified 2 groups (“pure” subtypes and recombinant forms) and resistance mutations (to PIs, NRTIs, and NNRTIs).

Categorical data assessed in the study were presented as proportions and frequencies and compared using the chi-square (χ^2^) test; in case of instability (for small sample sizes and/or for expected observation values < 5.0 in more than 1 cell—for four-field tables or in more than 20%—for arbitrary ones), Fisher’s exact test was used. DRM prevalence estimates were calculated with 95% confidence intervals (CIs). A statistically significant result was determined by *p* < 0.05.

Data analysis was performed using the R programming language (RStudio v.1.3.1093, Inc. Software, Boston, MA, USA) and STATISTICA v. 6.0 (StatSoft, Tulsa, OK, USA).

## 3. Results

### 3.1. Profile of the Study Cohort

A total of 79 HIV-1 sequences were analysed, obtained from ART-treated patients in the 2022–2024 period. The study cohort’s average duration time on ART treatment was 1101 days, equivalent to 3 years. The median age of participants was 40 years, ranging from 29 to 51 years. Of the cohort, 45 (56.96%) individuals were male and 34 (43.04%) were female. The main risk factor was sexual contact (67, 84.81%), followed by intravenous drug user (IDU) (10, 12.66%), and two cases of mother-to-child transmission (2.53%). Median (IQR) CD4 cell count was 435 (259.75–617.75) cells/mm and median (IQR) CD4 cell percentage was 20 (13–30.25 %). Median (IQR) HIV RNA was 11,020 (1770–44,670) copies/mL

### 3.2. Phylogenetic Analysis

The combined refined results of the primary identification of the HIV-1 genetic variants and phylogenetic analysis showed the following ratio: A6—57 (72.15%), CRF63—8 of sequences (10.13%), URFs—6 (7.59%), C—4 (5.06%), B—3 (3.8%), and CRF157—1 (1.27%) (Figure 1).

Unique recombinant forms were represented by different compositions of genovariants, such as BC, BG, CRF63B, and A6B.

### 3.3. Phylogenetic Clusters

The phylogenetic tree revealed 51 distinct clusters (with an intra-cluster genetic distance threshold of ≤1.5%). Among these, 15 (29.41%) clusters contained 31 sequences (17.61%) from the Far Eastern Federal District (Figure 2).

These 15 clusters were composed of 10 clusters of CRF63 (66.67%), 2 clusters of CRF02 (13.33%), and 1 cluster (6.67%) for each genetic variant: B, C, and G. Thus, the phylogenetic cluster analysis revealed a statistically significant difference in the number of clusters depending on the genetic variant: CRF63 vs. CRF02 (*p*-value = 0.003) and CRF63 vs. B/C/G (*p*-value < 0.001). And 31 sequences were composed of 21 sequences of CRF63 (67.74%), 4 sequences of CRF02 (12.90%), and 2 sequences (6.45%) for each genetic variant: B, C, and G.

A comparison of cluster sizes revealed that clusters of CRF63 were larger than those of other genetic variants. Thus, the maximum cluster size for CRF63 was seven sequences, compared to only two sequences for other genetic variants. In total, 10 CRF63 clusters were identified: one with seven sequences, one with five sequences, two with three sequences, and six with two sequences. Two CRF02 clusters with two sequences were also identified. And one cluster containing two sequences was found for each genetic variant: B, C, and G.

For most clusters, the modes of transmission were unknown. However, among people with known transmission modes, infection through sexual contact prevailed. The largest cluster, formed by seven HIV-1 CRF63 sequences, deserves special attention, as it includes multiple modes of transmission, including nosocomial infection and injection drug use.

As for the origin of the infection, 11 clusters (73.33%) included 27 people (87.09%) from the studied cities of the Far Eastern Federal District (Yakutsk, Khabarovsk, Yuzhno-Sakhalinsk, Vladivostok, Birobidzhan, Ulan-Ude). The remaining four clusters (0.27%) also included people from Siberian cities (Kemerovo, Novosibirsk, Novokuznetsk, Krasnoyarsk).

With an intra-cluster genetic distance threshold of ≤0.5%, the phylogenetic tree revealed 16 distinct clusters. Among these, five (31.25%) clusters contained 10 sequences (5.68%) from the Far Eastern Federal District (Figure 2). These five clusters were composed of CRF63 and included 10 people from the studied cities of the Far Eastern Federal District (Yakutsk, Birobidzhan, and Ulan-Ude).

Moreover, on the phylogenetic tree, the studied sequences of the genetic variant CRF63 formed multiple reliable clusters among themselves, which indicates multiple cases of the transmission of the virus of this variant within the country. The sequences of the recombinant form CRF02 formed several reliable phylogenetic clusters: (1) with sequences of African origin and (2) with sequences identified in the territory of the former Soviet Union (Kyrgyzstan, Tajikistan, Uzbekistan). Also, the sequences of CRF03 formed a reliable cluster with sequences mainly from Russia, and the cluster also included one sequence each from the UK and Spain. The only sequence CRF157 detected in the study, identified in Russia, was included in a reliable cluster with the reference sequences.

The sequences of the least characteristic for Russia recombinant forms CRF01 were included in a reliable cluster with sequences from Vietnam and China; CRF07—with sequences from China, and CRF11 and “pure” subtype A7—with sequences of African origin.

The studied sequences of subtype B formed several reliable phylogenetic clusters both among themselves and with sequences from the Commonwealth of Independent States (CIS) countries, as well as Europe (Great Britain, Germany) and Asia (Thailand). The studied sequences of subtype C also formed multiple reliable ones among themselves, and also entered a reliable cluster with sequences of African origin. The studied sequences of subtype G formed several reliable phylogenetic clusters: (1) with sequences from countries mainly in Eastern Europe, and (2) with sequences of Asian origin.

Additionally, an analysis was conducted for the most common genetic variant in Russia, A6. The phylogenetic tree revealed 25 distinct clusters. Among these, eight (32%) clusters contained 19 sequences from the Far Eastern Federal District and Moscow (Central Federal District) (Appendix A). The remaining clusters were found in other cities and regions of the Russian Federation, or the data (about cities in Russia) were unclear. Sub-subtype A6 sequences were also identified, forming several significant phylogenetic clusters with sequences from Ukraine and Estonia.

### 3.4. Analysis of Drug Resistance Mutations

To assess the prevalence of key drug resistance mutations (DRMs), we analysed PR-RT sequences (n = 79) targeting resistance to protease inhibitors (PIs), nucleoside reverse transcriptase inhibitors (NRTIs), and non-nucleoside reverse transcriptase inhibitors (NNRTIs). DRMs associated with NRTIs were the most frequently observed—55.7% (95% CI: 44.75–66.65%) (44/79), followed by those related to NNRTIs—27.85% (95% CI: 17.97–37.73%) (22/79) and PIs—6.33% (95% CI: 0.96–11.70%) (5/79).

The most commonly detected NRTI DRMs were A62V (43.04%, 34/79) and M184V (13.92%, 11/79); NNRTI DRMs were E138A (11.39%, 9/79), V106I (7.59%, 6/79), and H221Y (7.59%, 6/79); PI DRMs was M46I (3.8%, 3/79). It is worth noting that A62V and E138A are polymorphic mutations for HIV-1 subtype A6.

## 4. Discussion

This study is focused on the molecular epidemiology of HIV-1 genetic variants in Russia, using the Far Eastern Federal District as a case study. The study includes an analysis of HIV-1 transmission clusters in the region, as well as an examination of mutations leading to drug resistance. The results provide practical knowledge for improving HIV-1 prevention and treatment strategies in Russia and hold particular significance for vaccine development.

The demographic and epidemiological characteristics of the study population demonstrated a predominance of males and heterosexual transmission; the median age was 40 years, which generally coincides with such indicators both in individual regions of Russia and across the country as a whole [12,21,22].

Our analysis revealed the following distribution of HIV-1 genetic variants: the sub-subtype A6 virus was dominant in the Far Eastern Federal District, accounting for 72.15%, which is typical for HIV-1 in the Russian Federation and may be attributed to the “founder effect”, as this genetic variant was responsible for the rapid and widespread dissemination of HIV across the country in the late 1990s [8]. Despite the significant presence of sub-subtype A6, the region also exhibited high genetic diversity, including circulating and unique recombinant forms. The second most common variant was CRF63, accounting for 10.13%. This genetic variant of HIV-1 was identified in Siberia in the early 2010s, and its widespread distribution in the Far Eastern Federal District may be linked to the proximity of federal districts along with the “founder effect” [23]. The subtype B virus accounted for 3.8%, subtype C for 5.06% (which is rare in Russia), and one case of infection with a new recombinant form of HIV-1, identified in 2023 in the territory of Russia, in the Far Eastern Federal District, CRF157, was also detected. The share of recombinants was 7.59%, which corresponds to global trends towards an increase in recombinant form viruses on a global scale [2,5]. An analysis of additional sequences from the international Los Alamos National Laboratory database also identified the presence of CRF02, which entered Russia from Cameroon via Central Asian countries [24], CRF03, identified in the late 1990s in Kaliningrad (Russia) [25], subtype G, as well as sporadic detections of rare genetic variants of HIV-1 in Russia, such as CRF01, CRF07, CRF11, and A7, which are characteristic of Asian and African countries, respectively [9,26,27].

The phylogenetic cluster analysis identified 15 (or 5) clusters among the studied HIV-1 sequences. Among these, CRF63 formed a significantly larger proportion of the total clusters compared to any other non-A6 variant, suggesting more dynamic transmission patterns. It is noteworthy that these CRF63 clusters were formed by sequences obtained from the Siberian and the Far Eastern Federal Districts, indicating local transmission of the virus and underscoring the importance of national monitoring initiatives and studies of this nature.

Local transmission was also detected among the viruses of the HIV-1 genetic variants: CRF02, B, C, and G. At the same time, on the phylogenetic tree, the sequences of these genetic variants formed reliable clusters with sequences of diverse geographic origins: CRF02—from countries of the former USSR and Africa, B—from the CIS countries, Europe and Asia, C—from African countries, G—from Eastern European and Asian countries. This indicates a varied geographic origin and further spread of these genetic variants within our country. Sporadic cases of detection, without transmission clusters, were noted for the circulating recombinant forms CRF03 and CRF157, which were detected directly in Russia, as well as for CRF01, CRF07, and CRF11. For CRF03, limited introductions and reduced transmission efficiency may be attributed to the remoteness of the federal districts (Northwestern and Far Eastern), while for CRF157, a short circulation time may explain its sporadic presence. The sequences of recombinant forms CRF01 and CRF07 formed reliable phylogenetic clusters with sequences from Asian countries, which may be related to the recent popularisation of these tourist destinations.

Among the viruses of “pure” subtypes (predominantly HIV-1 sub-subtype A6), mutations associated with drug resistance to the NRTI class were prevalent. The primary mutations observed were A62V (43.04%) and M184V (13.92%). The former is a polymorphic mutation characteristic of HIV-1 sub-subtype A6, and its prevalence is linked to the dominance of this genetic variant. The M184V/I mutations confer resistance to 3TC/FTC, reducing vulnerability to these medications by more than 200 times [28,29,30], which are included in the first-line antiretroviral therapy regimen in Russia. This highlights the necessity for continuous monitoring of resistance trends to ensure the effectiveness of ART.

To date, several studies have been conducted on the development of anti-HIV vaccines, such as those in Thailand based on the CRF01 genetic variant and in South Africa focusing on clade C viruses, since these genetic variants are the most widespread in these regions [31].

Thus, when developing an HIV-1 vaccine for patients from Russia, it is essential to consider specific regional features of HIV-1, particularly the prevalence of sub-subtype A6. In 2019, an approach was proposed to create a stabilised HIV-1 envelope glycoprotein for vaccination strategies aimed at inducing broadly neutralising antibodies (bNAbs), based on the consensus sequence of all HIV-1 group M isolates [32]. This approach should probably also be considered in the context of HIV-1 sub-subtype A6 and other genetic variants that are most common in Russia. The results of this study indicate that viruses of genetic variants such as B and C, which are prevalent in Europe and America, and Africa, respectively, are also found in Russia [2].

Also, subtype-specific phenotypic differences, including coreceptor usage, replication fitness, disease progression, transmission biology, antigenicity, and mutational patterns, have been observed [33]. Even within subtypes, variations exist; for example, analysis of Pol and Env sequences has revealed mutations distinguishing A6 and A1 sequences, some in antibody-binding regions, potentially affecting immune response efficacy [34]. Subtype-specific signature amino acid residues, such as those in HIV-1 Tat protein (Indian HIV-1C), highlight the need for tailored vaccine design [35].

Thus, when assessing the effectiveness of vaccines under development, including those aimed at producing bNAbs, it is also important to use a panel of pseudoviruses that encompasses these genetic variants. Currently, no widely used panel includes the virus of sub-subtype A6 [36,37], despite its widespread distribution not only in Russia but also in European countries [9,27].

Additionally, based on the results of this study, it is important to emphasise the need for reliable epidemiological surveillance of CRF63 in the context of regional migration.

## 5. Conclusions

The results of this study highlight several important indicators for public health, particularly in the development of vaccines aimed at combating HIV infection.

Conducting molecular surveillance of HIV in the country is essential for obtaining timely data on HIV infection indicators, including the composition of genovariants and drug resistance mutations. This information is invaluable for prompt responses and is crucial for the development of new antiretroviral drugs and vaccines.

## Figures and Tables

**Figure 1 viruses-17-01568-f001:**
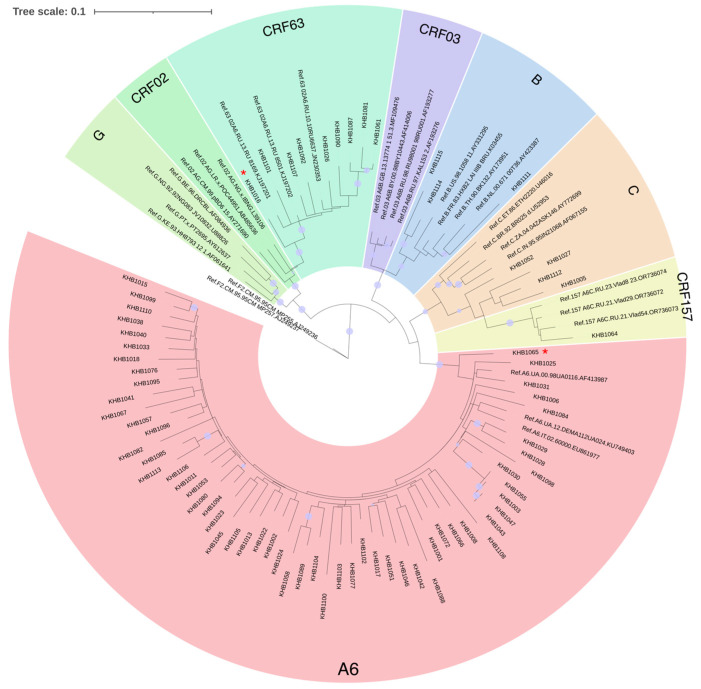
Results of phylogenetic analysis for the studied nucleotide sequences (n = 75). The sequences of viruses of unique recombinant forms are marked with a red asterisk. The purple dots indicate nodes with support above 95.

**Figure 2 viruses-17-01568-f002:**
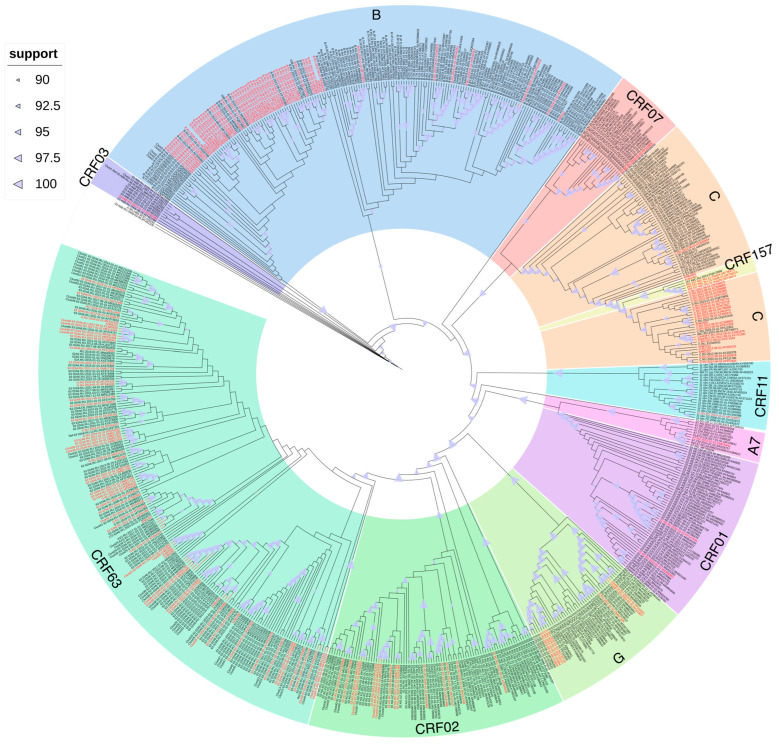
Results of cluster analysis of nucleotide sequences of viruses other than the A6 variant from the Far Eastern Federal District. Sequences from the Far Eastern Federal District of the Russian Federation are highlighted in red.

## Data Availability

All nucleotide sequences analysed in the study were deposited in the Los Alamos National Laboratory International Database.

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
