# Peer review of "An Assessment of Regional Genetic Diversity of HIV-1"

_viruses, 2025, doi:10.3390/v17121568_

Round 1

Reviewer 1 Report

Comments and Suggestions for Authors

General comments:

  1. What is the study cohort’s average duration time on ART treatment? This treatment data including viral load data would be useful for drug resistance analysis.
  2. Because samples were not representative due to the small study sample size and convenient samples used, statistical significance analysis might be biased. Simple descriptive statistics should be enough.
  3. When describing the transmission, A6 should be included in the analysis because it is the largest cluster circulating in the region.
  4. Study data was mainly limited to subtype distribution and transmission; no data was generated to support in vaccine development. Suggest removing the vaccine development from the manuscript title. In addition, no actual HIV vaccine is successfully developed or implemented so far, the detected subtypes in the city were not new and provide very limit value for vaccine development. However, it is ok to discuss their significance in the Discussion section.

Specific comments:

Introduction

Ln 58-60: please update the number of CRFs, the citation is out of date.

Method

2.1 How many sequences were downloaded from Los Alams and used in the study? What are their collection dates? These data are useful for transmission analysis.

Results

3.1 Ln 190: “usage” should be replaced by “User”, a common term for IDU.

3.2 this section might not be needed as the adjusted subtype distribution data was described in 3.3.

Fig 1 and Fig 2: name of subtypes are not identical as described in the text. Please have consistent naming/labeling.

Author Response

Thank you for your review of our manuscript. We appreciate the time and effort that you dedicated to providing feedback on our manuscript and are grateful for the insightful comments. We have answered each of your points below.

General comments:

Comments 1: What is the study cohort’s average duration time on ART treatment? This treatment data including viral load data would be useful for drug resistance analysis.

Response 1: The study cohort’s average duration time on ART treatment was 1101 days, equivalent to 3 years. The corresponding data have been updated to the text of the article. We have supplemented the text of the article with this treatment data (lines 196-197, 202).

Comments 2: Because samples were not representative due to the small study sample size and convenient samples used, statistical significance analysis might be biased. Simple descriptive statistics should be enough.

Response 2: Agree, we left a simple descriptive statistic of the study samples in the text of the article and excluded the statistical significance analysis (“Statistical analysis revealed statistically significant differences in the frequency of drug resistance mutations to NRTIs among those infected with viruses of “pure” HIV-1 subtypes compared to recombinants (p = 0.012).” from paragraph 3.5. Analysis of Drug Resistance Mutations and “The study revealed statistically significant differences in the nature of drug resistance between genetic variants” from paragraph 4. Discussion.

Comments 3: When describing the transmission, A6 should be included in the analysis because it is the largest cluster circulating in the region.

Response 3: Thank you for pointing this out. Initially, we planned to conduct a transmission cluster analysis for HIV-1 genetic variants that are less common than variant A6 (which is characteristic of Russia or the former Soviet Union) or are not characteristic of Russia at all, with the aim of identifying potential pathways for the introduction of these genetic variants. However, we also carried out an additional analysis for the A6 sub-subtype, in accordance with Your comments. The results of the analysis are presented in the text of the article and in the supplementary materials (lines 268-274).

Comments 4: Study data was mainly limited to subtype distribution and transmission; no data was generated to support in vaccine development. Suggest removing the vaccine development from the manuscript title. In addition, no actual HIV vaccine is successfully developed or implemented so far, the detected subtypes in the city were not new and provide very limit value for vaccine development. However, it is ok to discuss their significance in the Discussion section.

Response 4: Thank you for pointing this out. We removed the vaccine development from the manuscript title.

Specific comments:

Comments 1: Introduction. Ln 58-60: please update the number of CRFs, the citation is out of date.

Response 1: Thank you. We updated the number of CRFs (line 58).

Comments 2: Method. 2.1 How many sequences were downloaded from Los Alams and used in the study? What are their collection dates? These data are useful for transmission analysis.

Response 2: The total number of sequences downloaded from GenBank was 692 nucleotide sequences, 176 of which were from the Far East. Collection dates of sequences from the Far East ranged from 2012 to 2024.

The corresponding additions have been made to the text of the article (lines 164-166).

Comments 3: Results. 3.1 Ln 190: “usage” should be replaced by “User”, a common term for IDU.

Response 3: Thank you. We replaced it.

Comments 4: Results. 3.2 this section might not be needed as the adjusted subtype distribution data was described in 3.3.

Response 4: Agree, we combined these two paragraphs (3.2 and 3.3).

Comments 5: Fig 1 and Fig 2: name of subtypes are not identical as described in the text. Please have consistent naming/labeling.

Response 5: Thank you for pointing this out. We have agreed on the naming/labeling.

We also edited and improved the English language.

Reviewer 2 Report

Comments and Suggestions for Authors

In this manuscript, the authors described a molecular epidemiology study of HIV-1 in Far East region of Russia. The authors found out that circulating HIV-1 has high genetic diversity in this area and found several clusters using data from published HIV sequences in this region. Such data is valuable for readers who study HIV-1 global epidemiology and phylodynamics. I have the following comments and suggestions. 

1. Are these 79 patients that the authors reported naive patients or treatment-experienced patients? This will make a difference because treatment-experienced patients may have more resistance mutations.  

2. Please include the description of the ethical approval of this study. 

3. Why did the author exclude A6 from cluster analysis? Another comment is that when detecting genetic clusters in non-B HIV-1, it is often recommended to use different levels of cut-offs less than 1.5%. The authors should at least include an analysis using 0.5% as a cut-off.  

4. Line 306-307 should be included in the Results section, and the authors should expand this analysis to include more details.  

5. I encourage the authors to include more background data about HIV the study area in general.  

Author Response

Thank you for your review of our manuscript. We appreciate the time and effort that you dedicated to providing feedback on our manuscript and are grateful for the insightful comments. We have answered each of your points below.

General comments:

Comments 1: Are these 79 patients that the authors reported naive patients or treatment-experienced patients? This will make a difference because treatment-experienced patients may have more resistance mutations.

Response 1: These 79 patients are treatment-experienced, as noted in the text of the article (line 195).

Comments 2: Please include the description of the ethical approval of this study.

Response 2: Thank you for pointing this out. We included the description of the ethical approval of this study (lines 398-400).

Comments 3: Why did the author exclude A6 from cluster analysis? Another comment is that when detecting genetic clusters in non-B HIV-1, it is often recommended to use different levels of cut-offs less than 1.5%. The authors should at least include an analysis using 0.5% as a cut-off.

Response 3: Initially, we planned to conduct a transmission cluster analysis for HIV-1 genetic variants that are less common than variant A6 (which is characteristic of Russia or the former Soviet Union) or are not characteristic of Russia at all, with the aim of identifying potential pathways for the introduction of these genetic variants. However, we also carried out an additional analysis for the A6 sub-subtype.

And also agree with your comment and performed and included an analysis using 0.5% as a cut-off.

The results of the analysis are presented in the text of the article and in the supplementary materials (lines 214-216, 243-247 and 268-274).

Comments 4: Line 306-307 should be included in the Results section, and the authors should expand this analysis to include more details.

Response 4: Since this sentence (line 306-307: “No statistically significant differences were found in demographic characteristics and HIV-1 transmission characteristics depending on the HIV-1 genetic variant”) reflected the comparison results for the studied samples, whose demographic characteristics and route of infection were precisely known, but the samples were not representative due to the small study sample size and convenient samples used, we decided to limit ourselves to simple descriptive statistics.

At the same time, the statistical significance of the differences was reflected in the section on phylogenetic clusters, since in this case the total number of sequences analyzed was 708 (lines 224-225).

Comments 5: I encourage the authors to include more background data about HIV the study area in general.

Response 5: Agree, we included more background data about HIV the study area in general (lines 89-95).

Reviewer 3 Report

Comments and Suggestions for Authors This study assessed the genetic diversity of HIV-1 in the Far Eastern Federal District (FEFD) of Russia to inform effective anti-epidemic measures and vaccine development. The research involved analyzing 79 nucleotide sequences from patients in Khabarovsk city between 2022 and 2024, performing phylogenetic clustering using additional data from the Los Alamos database, and identifying drug resistance mutations (DRMs). The results confirmed the dominance of the sub-subtype A6 virus (72.15%), a characteristic feature of the Russian HIV epidemic due to a founder effect (pp. 1, 8). The second most common variant was CRF63_02A6 (10.13%). Phylogenetic cluster analysis revealed statistically significant differences in clustering patterns depending on the genetic variant, suggesting active local transmission, particularly for CRF63_02A6. Regarding drug resistance, DRMs to nucleoside reverse transcriptase inhibitors (NRTIs) were the most frequently observed (55.7%), with A62V and M184V being the most common specific mutations. The authors conclude that these specific regional features, particularly the prevalence of A6, must be considered when developing effective HIV-1 vaccines and treatment strategies tailored for the Russian population. Some points for improvement: 
  1. The abstract and methods state that sequences were obtained from "ART-treated patients" (pp. 1, 3). However, the median viral RNA level was quite high (11020 copies/mL) (p. 4). The authors should clarify if all patients were currently on a stable ART regimen at the time of sampling or if they had previous exposure to ART, which would explain the high prevalence of observed drug resistance mutations (DRMs) (p. 7). This clarification is essential for interpreting the DRM data accurately.
  2. The authors mention "statistically significant differences" in cluster numbers and DRM frequencies between genetic variants (pp. 1, 6-7). It would improve the clarity and impact of the results section to explicitly list the calculated p-values alongside these claims within the main text of the results section, rather than just in the abstract or discussion (pp. 1, 6).
  3. The link between the molecular data and "vaccine development" could be stronger in the discussion. While the manuscript correctly highlights the need to use a representative panel including the A6 subtype, expanding on how the specific A6 sequence characteristics (beyond simple prevalence) might influence immunogen design would be beneficial (p. 9).

Author Response

Thank you for your review of our manuscript. We appreciate the time and effort that you dedicated to providing feedback on our manuscript and are grateful for the insightful comments. We have answered each of your points below.

General comments:

Comments 1: The abstract and methods state that sequences were obtained from "ART-treated patients" (pp. 1, 3). However, the median viral RNA level was quite high (11020 copies/mL) (p. 4). The authors should clarify if all patients were currently on a stable ART regimen at the time of sampling or if they had previous exposure to ART, which would explain the high prevalence of observed drug resistance mutations (DRMs) (p. 7). This clarification is essential for interpreting the DRM data accurately.

Response 1: Thank you for pointing this out. Most patients were on a stable ART regimen, with the average duration of ART being approximately 1,101 days (approximately three years), which we added to the text of the article (lines 195-197 ). However, some patients experienced adverse events, or the treatment regimen was optimised to improve adherence. Therefore, the duration of therapy combined with non-compliance with the prescribed medication regimen, may have contributed to the persistently high viral load. The low proportion of patients with suppressed viral load in the Far Eastern Federal District (among all regions of Russia) was also noted in a previous study assessing HIV infection trends in Russia, further underscoring the importance of conducting such studies in the region. All necessary additions have been made to the text of the article (lines 92-95).

Comments 2: The authors mention "statistically significant differences" in cluster numbers and DRM frequencies between genetic variants (pp. 1, 6-7). It would improve the clarity and impact of the results section to explicitly list the calculated p-values alongside these claims within the main text of the results section, rather than just in the abstract or discussion (pp. 1, 6).

Response 2: Thank you for pointing this out. Since this sentence (line 306-307: “No statistically significant differences were found in demographic characteristics and HIV-1 transmission characteristics depending on the HIV-1 genetic variant”) reflected the comparison results for the studied samples, whose demographic characteristics and route of infection were precisely known, but the samples were not representative due to the small study sample size, we decided to limit ourselves to simple descriptive statistics.

At the same time, the statistical significance of the differences was reflected in the section on phylogenetic clusters, since in this case the total number of sequences analyzed was 708. For these differences, p-values were reported in the main text in the Results section (lines 224-225).

Comments 3: The link between the molecular data and "vaccine development" could be stronger in the discussion. While the manuscript correctly highlights the need to use a representative panel including the A6 subtype, expanding on how the specific A6 sequence characteristics (beyond simple prevalence) might influence immunogen design would be beneficial (p. 9).

Response 3: Agree. We supplemented the Discussion section in accordance with your comment (lines 359-365).

Round 2

Reviewer 2 Report

Comments and Suggestions for Authors

The authors have addressed my comments

Reviewer 3 Report

Comments and Suggestions for Authors

The article has been revised/corrected according to all the reviewers' comments. It looks excellent now. I recommend its publication as is.